# B7-H3 Associates with IMPDH2 and Regulates Cancer Cell Survival

**DOI:** 10.3390/cancers15133530

**Published:** 2023-07-07

**Authors:** Salwa Alhamad, Yassmin Elmasry, Isabel Uwagboe, Elena Chekmeneva, Caroline Sands, Benjamin W. Cooper, Stephane Camuzeaux, Ash Salam, Maddy Parsons

**Affiliations:** 1Randall Centre for Cell and Molecular Biophysics, King’s College London, Guys Campus, New Hunts House, London SE1 1UL, UK; 2Department of Biology, College of Science, Imam Abdulrahman Bin Faisal University, P.O. Box 1982, Dammam 31441, Saudi Arabia; 3National Phenome Centre, Section of Bioanalytical Chemistry, Department of Metabolism, Digestion and Reproduction, Imperial College London, Hammersmith Hospital Campus, IRDB Building, 5th Floor, Du Cane Road, London W12 0NN, UKash.salam@imperial.ac.uk (A.S.)

**Keywords:** B7-H3, lung cancer, IMPDH2, proliferation, invasion, oxidative stress, cell–cell adhesion, rods and rings, cell survival, chemoresistance

## Abstract

**Simple Summary:**

Lung cancer is one of the most common cancers in the world, and despite improvements in treatment, it remains a significant health burden. B7-H3 is a member of the B7 immune checkpoint family of membrane proteins and shows increased expression in lung cancers; however, the functional consequences of this to cancer cell signalling remain unclear. Here, we show that B7-H3 forms a complex with the metabolic enzyme IMPDH2, and this protects cancer cells from oxidative stress and resulting apoptosis triggered by chemotherapy. The removal of B7-H3 from cancer cells in 2D results in sensitisation to chemotherapeutic stress and reduces cell viability in an IMPDH2-dependent manner. However, the loss of B7-H3 in 3D spheroid models results in enhanced IMPDH2-dependent spheroid growth, likely due to the weakening of cell–cell adhesion. These findings demonstrate that B7-H3 functionally contributes to metabolic signalling in lung cancer cells in the absence of immune cell interactions.

**Abstract:**

Lung cancer is one of the most common cancers worldwide, and despite improvements in treatment regimens, patient prognosis remains poor. Lung adenocarcinomas develop from the lung epithelia and understanding how specific genetic and environmental factors lead to oncogenic transformation in these cells is of great importance to define the pathways that contribute to tumorigenesis. The recent rise in the use of immunotherapy to treat different cancers has prompted the exploration of immune modulators in tumour cells that may provide new targets to manipulate this process. Of these, the B7 family of cell surface receptors, which includes PD-1, is of particular interest due to its role in modulating immune cell responses within the tumour microenvironment. B7-H3 (CD276) is one family member that is upregulated in many cancer types and suggested to contribute to tumour–immune interactions. However, the function and ligand(s) for this receptor in normal lung epithelia and the mechanisms through which the overexpression of B7-H3 regulate cancer progression in the absence of immune cell interactions remain unclear. Here, we present evidence that B7-H3 is associated with one of the key rate-limiting metabolic enzymes IMPDH2, and the localisation of this complex is altered in human lung cancer cells that express high levels of B7-H3. Mechanistically, the IMPDH2:B7-H3 complex provides a protective role in cancer cells to escape oxidative stress triggered by chemotherapy, thus leading to cell survival. We further demonstrate that the loss of B7-H3 in cancer cells has no effect on growth or migration in 2D but promotes the expansion of 3D spheroids in an IMPDH2-dependent manner. These findings provide new insights into the B7-H3 function in the metabolic homeostasis of normal and transformed lung cancer cells, and whilst this molecule remains an interesting target for immunotherapy, these findings caution against the use of anti-B7-H3 therapies in certain clinical settings.

## 1. Introduction

Lung cancer is the most common global type of cancer and most common cause of cancer fatality [1]. Although cigarette smoking remains the main risk factor of lung cancer [2], environmental factors, exposure to carcinogens, and chronic lung disease also contribute to disease onset. Despite recent improvements in survival for many cancers, lung cancer has a relatively poor 5-year survival rate [3]. Treatments include chemotherapy, surgery, radiation, targeted therapy, and immunotherapy [4]. Targeted therapies and immunotherapy have recently shown promising results in lung cancer treatment [5]; however, many patients remain refractory to treatment.

Most solid tumours arise from the epithelium and understanding this tissue type aides in determining ways in which cancer cells deviate from homeostasis to become malignant [6]. Epithelial integrity and homeostatic cell numbers are maintained by balancing cell death with dividing cells, which is essential to survival, and many cellular mechanisms ensure that these processes are maintained during growth and in response to damage [7]. Tumorigenesis is the acquisition of malignant properties in normal cells: uncontrolled growth, metastasis, evasion of immunosurveillance and apoptosis, and dysregulated metabolism and epigenetics are regarded as the hallmarks of cancer [8].

Cancer cells increase in biosynthetic processes and energy production to support proliferation and cell growth [9]. Increased nutrient uptake via phagocytosis or micropinocytosis can occur alongside an increased demand for nitrogen derivatives and their nucleotide conversions (pyrimidines, purines). Changes in metabolic-driven genes can occur via methylation, acetylation, or metabolic interaction with the tumour microenvironment (TME). This assists tumour growth through the exchange of aminoamides, nutrients, or secreted growth factors, or through environmental conditions such as redox stress, hypoxia, and alterations to the extracellular matrix and cell–cell interactions [10,11]. Purines are the most abundant metabolic substrates providing building blocks for DNA and RNA as well as energy and cofactors to support cell survival and proliferation. Therefore, purines and their derivatives play a key role in most biological processes, including immune responses [12]. High levels of purine metabolites have been discovered in tumour cells, which has led to the development of the earliest anti-tumour drugs to treat cancers (purine anti-metabolites) by blocking both DNA synthesis and cell growth [13]. Purines and the de novo purine biosynthetic pathway enzymes are enhanced in tumour cells [14].

IMPDH is the rate-limiting enzyme in the de novo biosynthesis of purine nucleotides that is essential for DNA synthesis [15]. IMPDH is involved with cell growth, differentiation, and malignant transformation [16]. IMPDH has two isoforms, IMPDH1 and IMPDH2, which are encoded by different genes with 84% of their amino acid sequence being conserved [17]. IMPDH2 is highly expressed in malignant cells, upregulated during proliferation, and can mediate chemoresistance [18,19,20]. IMPDH2 assembles into filamentous linear or circular micron-scale structures referred to as rods and rings (RRs) [21,22], which are correlated with a deficiency in GMP/GTP synthesis and decreased IMPDH2 activity [21]. IMPDH inhibitors such as mycophenolic acid (MPA) or ribavirin induce the formation of RRs [23,24]. However, recent evidence suggests that IMPDH2 point mutations that promote or block polymerisation do not affect its activity and both active and inactive forms of IMPDH2 can aggregate into filaments [25]; however, the presence of active forms of IMPDH2 in RRs in a cellular context remains unclear. RR remains were initially considered to be a cellular response to increased guanine nucleotide synthesis [26] and have subsequently been shown to be important for maintaining normal cell proliferation and GTP homeostasis [27]. However, the role and regulation of IMPDH2 in RRs remains unclear, in both normal epithelial and cancer cells [28].

Uncontrolled cell growth in cancer can drive increased metabolism and elevate basal levels of reactive oxygen species (ROS) compared to normal cells. As a result, cancer cells are constantly under oxidative stress [29]. ROS are important signal mediators in the tumour microenvironment involved in T cell and natural killer cell activation and the subsequent detection/destruction of transformed cells. However, cancer cells can induce tumour-promoting immune cells causing chronic inflammation that can promote cancer progression [30]. This positive feed-back loop occurs between oxidative stress and inflammatory mediators, shaping the outcome of anti-tumour immune responses [31]. Cancer cells and immunosuppressive cells use ROS in the TME along with other mechanisms to create immune tolerance to tumours [32,33,34]. B7 family members have been shown to exert both positive and negative roles in modulating tumour immune cell responses [35,36]. One family member, B7-H3 (CD276), is overexpressed in several tumour types, including non-small-cell lung cancer (NSCLC), ovarian, and prostate cancer, and its expression is associated with poor treatment outcomes and survival [37].

B7-H3 has been reported to have both co-stimulatory and co-inhibitory roles in different tumour microenvironments. B7-H3 was first identified as a positive regulator of T-cells when induced in murine lymphomas via CD8^+^ T and natural killer (NK) cell activation [38]. In a murine mastocytoma model, B7-H3 caused tumour regression through cytotoxic T cell induction [39]. Similar results were observed in murine models of colorectal carcinomas, suggesting that the enrichment of B7-H3 leads to an anti-tumour immune response [40]. However, more recent work has shown that B7-H3 induces a strong immune-evasive effect. The expression of B7-H3 is negatively correlated with circulating CD8^+^ tumour-infiltrating lymphocytes, suggesting a role in tumour evasion in hypopharyngeal squamous cell carcinoma and osteosarcoma [41,42]. In clear-cell RCC [43] and NSCLC [44], the expression of B7-H3 is correlated with circulating Tregs that suppress inflammation to promote the secretion of cytokines and growth factors, leading to tumour progression. B7-H3 inhibits the proliferation of both CD4^+^ and CD8^+^ T-cells and decreases IL-2 and IFN-γ production [45]; however, the ligand(s) for B7-H3 that mediate these biological effects remain unknown. The roles for B7-H3 in the tumour compartment outside of immune regulation are less well understood, as are the functions of this receptor in normal epithelia. B7-H3 has been implicated in the control of tumour epithelial–mesenchymal transition (EMT) through a Jak2/STAT3/Slug-dependent mechanism [46]. The downregulation of B7-H3 expression sensitises breast cancer cells to AKT/mTOR inhibitors and reduces glycolytic capacity [47]. These studies suggest that B7-H3 has non-immunological roles in cancer and could contribute to cellular metabolism and sensitivity to chemotherapeutic compounds. Indeed, recent studies found that B7-H3 depletion in melanoma cells increased in vivo and in vitro sensitivity to cisplatin and dacarbazine in parallel with a reduction in p38 MAPK phosphorylation [48].

Here, we aimed to understand B7-H3 function in normal lung epithelial cells compared with lung adenocarcinoma cells in the absence of interactions with the immune compartment. We demonstrate that the subcellular localisation of B7-H3 is altered in lung cancer cells compared to normal lung epithelia and assembles at cell–cell adhesions in the latter. B7-H3 forms a complex with the metabolic enzyme IMPDH2, leading to enhanced GTP synthesis and the protection of cells against triggers of oxidative stress. Surprisingly, B7-H3 depletion from cancer cells does not affect growth or migration in 2D but promotes invasion in 3D models in an IMPDH2-dependent manner. These findings provide new insights into B7-H3 function in the metabolic homeostasis of normal and transformed lung cancer cells and caution against the use of B7-H3-depleting agents in the treatment of solid lung cancers.

## 2. Materials and Methods

### 2.1. Antibodies and Reagents

Anti-B7-H3 (ThermoFisher, Loughborough, UK (IP), AF1027 BD Bioscience, Wokingham, UK (WB), and MAB1027 Santa Cruz, Heidelberg, Germany (IF)), anti-EEA1 (Cell Signaling, Leiden, The Netherlands), anti-HSC70 (Santa Cruz), anti-IMPDH2 (Proteintech), anti-Rab10 (Abcam, Cambridge, UK), anti-Rab11 (Cell Signaling), and HTII-280 AT2 (terrace Biotech, San Francisco, USA) were used. IgG negative control was used for IP and obtained from DAKO. DAPI (Sigma-Aldrich, Gillingham, UK) was used as nuclear stain for immunofluorescence. LysoTracker™ Deep Red was used to stain lysosomes (ThermoFisher). Anti-mouse-HRP, anti-rabbit-HRP, and anti-goat-HRP were from DAKO; anti-mouse-568, anti-rabbit-568, anti-rabbbit-488, and phalloidin-488 and -647 were all obtained from Invitrogen. Other reagents used were CellROX™ Deep Red Reagent (for oxidative stress detection) (ThermoFisher). CellEvent™ Caspase-3/7 Green Detection Reagent (ThermoFisher), Cisplatin (Cambridge Bioscience, Cambridge, UK), and Mycophenolic acid (MPA; IMPDH inhibitor) (Sigma-Aldrich).

### 2.2. Plasmids, siRNA, and Primers

B7-H3-GFP, a c-terminally tagged transient expression plasmid, was generated by cloning the full-length human B7-H3 into GFP-N1 (Clontech) using Nhe1 and Kpn1 restriction sites. CD63-GFP, transient full-length human CD63, was a gift from Fedor Berditchevski (University of Birmingham). B7-H3 CRISPR constructs targeting CACGGCTCTGTCACCATCAC or TTGATGTGCACAGCGTCCTG were GFP-tagged and were obtained from GenScript. B7-H3 and IMPDH2 siRNA were used to obtain acute knockdown. All siRNA SMARTpool or individual oligos and non-targeting control siRNA pools were purchased from Horizon Discovery plc, Cambridge, UK.

### 2.3. Cell Culture

16HBE normal human bronchial epithelial cells, adenocarcinoma human alveolar basal epithelial cells A549, and bronchioalveolar carcinoma cell line NCI-H358 (all purchased from ATCC) were used in this study. Cells were grown in Minimum Essential Media (MEM) for 16HBE cells, Dulbecco’s modified eagle medium (DMEM) for A549, and RPM I-1640 for the NCI-H358 cells. Media were supplemented with 10% *v*/*v* foetal calf serum, 2 mM L-Glutamine, 100 units/mL penicillin, and 0.1 mg/mL streptomycin. B7-H3 CRISPR cell lines were established via transient transfection of A549 and H358 parental cells with the CD276 CRISPR Guide RNA vector carrying a GFP tag. The GFP-tagged cells were sorted 24 h post-transfection using flow cytometry to obtain a homogenous but non-clonal cell population. 16HBE cells did not respond to any of the B7-H3 CRISPR knockdowns and thus B7-H3 siRNA was used. Cells presenting the lowest B7-H3 expression level were chosen for further experiments. B7-H3 depletion was validated via sequencing, qPCR, Western blot, and immunofluorescence. Cell lines were treated with MPA to block IMPDH2 activity (1 µM for 24 and 48 h); methanol was used as a control.

### 2.4. Immunoprecipitation

16HBE, A549, and H358 cells were treated with or without MPA and lysed in 500 μL of IP Lysis buffer (50 mM Tris Base, 150 mM NaCl, 1 mM EDTA, 50 mM Sodium fluoride, 1% NP-40, 1% Triton X-100, protease and phosphatase inhibitors). Lysates were then rotated for 15 min at 4 °C and incubated with A/G agarose affinity matrix suspension. To pre-clear the lysates, the latter were incubated with A/G beads at 4 °C on a rotator and then centrifuged at 2000× *g* at 4 °C. The supernatant was split into tubes containing 3 µg of B7-H3, IMPDH2, or control IgG antibody and left tumbling overnight at 4 °C. A total of 100 ul of washed protein A/G bead slurry was added to the antibody/lysates mixture and tumbled for 2 h at 4 °C. Samples were then spun down at 2000× *g* for 3 min at 4 °C, the supernatant was discarded, and the beads were washed. Beads were boiled with 50 μL of 2× sample buffer containing (1:200) β-mercaptoethanol for 10 min at 95 °C and samples were loaded on a 10% SDS PAGE gel and subjected to immunoblotting using the specified antibodies.

### 2.5. Western Blotting

Cells were lysed in RIPA buffer (pH 7.4, 10 mM Tris Base, 150 mM NaCl, 1 mM EDTA, 1% Triton X- 100). β-mercaptoethanol was added to lysates in a 1:200 ratio. Lysates were subjected to SDS-PAGE and blotted using PVDF membrane. Blots were blocked and probed using 5% Bovine Serum Albumin in TBS-0.1% Tween. Proteins were detected using an ECL chemiluminescence kit (BioRad) and directly imaged using the BioRad imager digital imaging system. PNGase F was used to cleave all N-linked oligosaccharides from B7-H3. After cell lysates were collected, 20 µg of glycoprotein, 1× Glycoprotein Denaturing Buffer, and water were combined and denatured at 95 °C for 10 min. This was followed by the addition of 1× GlycoBuffer 2, 1% NP-40, PNGase F, and water. The reaction mix was incubated for 1 h at 37 °C before Western blot analysis.

### 2.6. Metabolite Extraction

In a 15 cm^2^ dish, 2 × 10^6^ 16HBE parental and siRNA (control and B7-H3) transfected cells and A549 and H358 parental and CRISPR lines were plated, and they were left to adhere overnight. Parental lines were either vehicle- or MPA-treated for 48 h. Media were then aspirated from the cells and the culture plates were washed twice with 20 mL of ice-cold 150 mM Ammonium Acetate, ensuring that it was completely removed from the plates after each wash. The dry plates were then incubated on dry ice for 2 min to quench the metabolome. Next, 2.2 mL of ice-cold methanol was added, and the cells were scraped and transferred to a pre-cooled 15 mL Falcon tube and left to rotate for 10 min at 4 °C. Then, 2.2 mL of LC-MS-grade H_2_O was added to the tube and mixed thoroughly via vortexing. The mixture was centrifuged at 1550 RPM for 45 min at 4 °C. A total of 4 mL of supernatant was then transferred to a clean 15 mL Falcon tube and dried under nitrogen. An aliquot of 1 mL of MeOH:H_2_O 1:1 was added to the dried cell extract in the tube and vortexed thoroughly. Afterward, the reconstituted extract was transferred to an Eppendorf tube and dried once more before storing it at -80 °C. Prior to the LC-MS analysis, the extracts were reconstituted in 100 µL of method-specific diluent, as described in the next section.

### 2.7. UHPLC-MS Metabolic Profiling

LC-MS metabolic profiling of cell extracts was carried out at the National Phenome Centre (Imperial College London) using hydrophilic interaction liquid chromatography assay in negative ionisation mode (HILIC-), developed for the analysis of polar cellular metabolites.

#### 2.7.1. Materials and Reagents

All solvents and mobile phase additives were LC-MS-grade. Acetonitrile (CHROMASOLV LC-MS) was purchased from Scientific Laboratory Supplies (Wilford, UK), methanol and water were purchased from Fisher Scientific (Leicestershire, UK), ammonium bicarbonate was obtained from Fluorochem (Hadfield, UK), and ammonium hydroxide was from Sigma-Aldrich (Gillingham, UK). All authentic chemical standards used for the annotation of metabolites were of analytical or higher grade and were obtained from various suppliers (Sigma-Aldrich (Gillingham, UK), VWR (Leicestershire, UK), and Fisher Scientific (Leicestershire, UK)). They were accurately weighed and dissolved in appropriate volumes of LC-MS-grade water to obtain individual stock solutions at concentrations of 1 mg/mL. Appropriate amounts of each stock solution were mixed to obtain four standard mixtures at concentrations of 25 µg/mL in water that were further prepared for HILIC analysis, as described below. Stable isotope-labelled internal standards AMP-^15^N_5_, ADP-^15^N_5_, and ATP-^13^C_10_ were purchased from Sigma-Aldrich (Gillingham, UK) and prepared in a mixture at a concentration of 48 µg/mL.

#### 2.7.2. Sample Preparation for HILIC Analysis

The dried cell extracts (n = 20) prepared as described in the previous section were reconstituted in 100 µL of LC-MS-method-specific diluent consisting of 1:1:2 H_2_O/methanol/acetonitrile solvent mixture. The aqueous part of the diluent incorporated stable isotope-labelled standards that were diluted in the diluent to a final concentration of 3 µg/mL. The reconstituted cell extracts were centrifuged at 3486× *g* for 10 mins at 4 °C (Eppendorf 5430R centrifuge) in Eppendorf tubes, and the supernatants were carefully transferred to LC vials. Aliquots of 10 µL of each reconstituted cell extract were mixed to generate a pooled study reference (SR) sample analysed along with the study samples.

Authentic standard mixtures were prepared for the analysis by adding one volume of methanol and two volumes of acetonitrile to one volume of aqueous 25 µg/mL solutions of the four mixtures (final concentration of the standards of 6.25 µg/mL).

#### 2.7.3. UHPLC-MS Analysis

Metabolic profiling experiments were performed using Acquity UPLC instruments coupled to Xevo G2-S QTOF mass spectrometers (Waters Corp., Manchester, UK) via a Z-spray electrospray ionisation (ESI) source in negative ionisation mode. The HILIC chromatographic retention and separation of polar cellular metabolites was conducted using a zwitterionic, sub 2 µm, Atlantis Premier BEH Z-HILIC column (1.7 µm, 2.1 × 150 mm, Waters Corporation, Milford, CT, USA) thermostatted at 40 °C. Mobile phase A was 5 mM ammonium bicarbonate in water adjusted to a pH of 9 with 4M ammonium hydroxide solution. Neat acetonitrile was used as mobile phase B. Both mobile phase A and B were filtered through 0.22 µm PTFE filters (Pall Europe, Portsmouth, UK) to avoid undesired particle precipitation in the chromatographic column and LC system. The initial conditions were 20:80 A:B at a flow rate of 0.4 mL/min. To elute the retained metabolites from the column, mobile phase A content was linearly increased to 50% in 8 min and further to 60% in 0.1 min, where it was maintained for 0.9 min to wash out all sample material from the column. Afterward, the column was returned to the initial conditions of 20% A in 1.4 min for equilibration in the next 5.2 min. The total run time was 15 min and the injection volume was 2 µL.

The mass spectrometry parameters were set as follows: capillary voltage 2.4 kV, sample cone voltage 20 V, source offset 80 V, source temperature 150 °C, desolvation temperature 600 °C, desolvation gas flow 1200 L/h, and cone gas flow 150 L/h. Centroid data were collected in sensitivity mode with dynamic range extension within a 50-1200 *m*/*z* range with a scan time of 0.07 s. For mass accuracy, lockspray mass correction was performed using a 200 pg/μL leucine enkephalin solution (*m*/*z* 554.2615 in ESI-) in 1:1 water/acetonitrile (with 0.1% formic acid) solution at a flow rate of 15 μL/min. Lockmass scans were collected every 60 s and averaged over 4 scans.

#### 2.7.4. Run Order and Quality Control

All reconstituted cell extracts were acquired in triplicate in a randomised order, and triplicate injections of the pooled SR sample were acquired at the beginning and the end of the study sample analysis to support the analytical quality assessment of the data. Standard mixtures were run before injecting the cell samples and bookending the run to account for any retention time shifts.

#### 2.7.5. Metabolite Annotation and Data Processing

The signals of the metabolites of interest detected in the cell extracts were annotated by matching them to authentic chemical standards and analysed as four mixtures, based on the retention time alignment in combination with *m*/*z* and MS/MS data matching. The signals of the annotated metabolites and labelled internal standards were integrated using TargetLynx Software v4.2 (Waters). Peak areas of metabolites were normalised according to the peak area of the closest in retention time internal standard, and precision (relative standard deviation, RSD) was calculated across triplicate injections of each sample. Normalised peak area values passing the threshold of RSD < 30% were used for further statistical analyses.

### 2.8. Immunofluorescence

Cells were plated on 13 mm coverslips and fixed after specified times or treatments using 4%PFA in PBS for 15 min at room temperature followed by washing three times with PBS. Cells were then permeabilised using 0.25% TritonX-100 in PBS for 10 min and washed again three times with PBS. Following permeabilisation, coverslips were blocked with 3% BSA/PBST for 1 h at room temperature. The incubation of cells with primary antibody diluted in 3% BSA/PBST was then carried out in a dark chamber at 4 °C overnight. Cells were incubated with secondary antibodies and DAPI diluted in 3% BSA/PBST for 2 h at room temperature. Coverslips were mounted on slides using FluorSave mounting media (Calbiochem).

### 2.9. Confocal Microscopy

The slides were then analysed using a A1R laser scanning confocal microscope (Nikon) with a 60×/1.4Plan-APOCHROMAT oil immersion objective. Images were acquired in ND2 format, exported as TIFFs, and analysed using Image J. Confocal microscopy images of fixed cells were acquired using a Nikon A1R inverted confocal microscope (Nikon Instruments UK) with an environmental chamber maintained at 37 °C/5% CO_2_. Images were taken using a 20× or 60× Plan Fluor oil immersion objective (numerical aperture of 0.45 and 1.4, respectively). Excitation wavelengths of 488 nm (diode laser), 561 nm (diode laser), or 640 nm (diode laser) were used. Images were acquired using NIS-Elements imaging software (v4) and were saved in Nikon Elements in ND2 format. Image processing was carried out using Fiji processing software (v4).

### 2.10. Measurement of Membrane and Cytoplasmic IMPDH2 Intensity

The membrane intensity of IMPDH2 staining was analysed using ImageJ. A region of interest was drawn around a single cell, including the visible membrane boundary, using phalloidin staining as a guide. Eight ROIs were selected for each field of view and grey values were measured. To measure the IMPDH2 intensity in the cytoplasm, the ROI of each selection was reduced on all sides of the cell through the use of the enlarge option by −7 pixels to remove the membrane contribution. The grey value of the smaller cytoplasmic ROI was then measured. Values were exported to excel and the percentage of the total signal in the membrane was calculated by dividing the smaller value (cytoplasm) by the larger one (membrane + cytoplasm). GraphPad Prism was used for statistical analysis.

### 2.11. Oxidative Stress Detection

16HBE parental, siRNA control and B7-H3 siRNA, A549 and H358 parental, or B7-H3 CRISPR cells were plated onto a 96-well plate and left to adhere overnight. Cells were then washed with PBS, media were changed to either complete media (control), complete media supplemented with MPA for parental lines only, or media without FBS (serum staved), and they were left to incubate for 48 h. A fluorogenic probe for measuring cellular oxidative stress and CellROX Deep Red reagent was added to the cells at 5 µM concentration together with DAPI to normalise the oxidative stress fluorescence to cell number. Cells were washed three times with PBS and imaged using an EVOS microscope. Fluorescence intensity was measured using Fiji software (v4).

### 2.12. Chemoresistance and Survival Assay

A549 and H358 parental and B7-H3 knockdown cells were used in this experiment in different conditions, cells were either untreated, MPA-treated, or starved for 48 h, and each condition was supplemented with 5 µM of either DMSO as the vehicle control or Cisplatin to trigger cell apoptosis. After 48 h, growth media were removed, and CellEvent^TM^ Caspase-3/7 Green Detection Reagent (ThermoFisher) was added to wells at 0.4 µM together with DAPI for 30 min as per the manufacturer’s protocol. Cells were then imaged using the GFP and DAPI channels via an EVOS microscope. The fluorescence was measured, and the number of cells was counted, both using Fiji software (v4).

### 2.13. Hanging Drop Spheroid Assay and Immunostaining

To generate hanging drops for spheroids, methylcellulose stock solution was made by pre-heating 250 mL DMEM to 60 °C and using pre-heated DMEM to dissolve 6 g of autoclaved methylcellulose powder. Then, the solution was stirred for 20 min at room temperature. An additional 250 mL DMEM was added, and the solution was stirred at 4 °C overnight. Methylcellulose solution was then cleared via centrifugation in 50 mL Falcon tubes at 5000× *g* at room temperature for 2 h. Cleared methylcellulose solution was transferred to fresh 50 mL Falcon tubes and stored at 4 °C. In 2.25 mL of DMEM or RPMI media, respectively, containing 0.5% FBS, 75 × 10^3^ H358 parental and B7-H3 CRISPR cells and 50 × 10^3^ A549 parental and B7-H3 CRISPR cells were resuspended. A total of 750 µL of methylcellulose was then added to the cell suspension. A total of 20 µL aliquots of the cell suspension was dropped onto the lid of a square Petri dish and 10 mL of PBS was added to the bottom of the dish. The lid containing the drops was carefully inverted and placed over the PBS. The hanging drops were incubated for 48 h to allow spheroids to form. To perform the invasion assay, a collagen mix containing 2 mg/mL collagen (Rat Tail Type I collagen), 20 mM Hepes, 10 mM fibronectin, 17.5 mM NaOH, and OptiMEM was made to embed the spheroids. A total of 100 µL of the collagen mix was added into each well of a 96-well plate. Each spheroid was removed from the hanging drop and dropped into each well before the collagen had set. The gels were then left to set at 37 °C for 1 h. Once the collagen was set, 100 µL of media was added into each well. Phase- contrast images of the spheroids post-embedding were acquired using an Evos FL Auto 2 fluorescent microscope (ThermoFisher) and a 3.2 MP CMOS camera with 10x objective at 0 and 24 h. Cell invasion was quantified using Fiji software. The acquired images (.TIFF format) were imported using Fiji, a region of interest was manually drawn around each spheroid, and the area was quantified using the software. Regions of interest were drawn around the entire spheroid (including single peripheral invading cells). Spheroid cell expansion was quantified by dividing the spheroid area at 24 h by the spheroid area measured at 0 h post-embedding. MPA was diluted in media and added on top of the embedded spheroids. Spheroids were fixed in 4% PFA for 2 h at 37 °C. Samples were washed thrice for 10 min in PBS followed by permeabilisation using 0.25% Triton X-100 in PBS (PBST, *v*/*v*) for 30 min at room temperature. Samples were washed for 10 min before blocking with 5% BSA in PBST for 1 h at room temperature on a rocker. DAPI and IMPDH2 antibodies and Phalloidin 488 diluted in blocking buffer were added to each embedded spheroid and left to stain on the rocker overnight. Samples were washed for 10 min three times before the final addition of 100 µL PBS, and stored at 4 °C covered in foil until imaging. Images were acquired using the confocal microscope and analysed using QuPath software (v0.4.3).

### 2.14. Human Lung PCLS Sectioning

Tissue slices were created following our previous protocols [49]. Distal human lung tissue samples were obtained through the Royal Brompton Hospital Biobank after ethics approval with the NRES reference 20/SC/0142, and stored in RPMI (supplemented with 1% (*v*/*v*) penicillin-streptomycin and gentamycin, both at 10 mg/mL) at 4 °C until further processing. All cell culture reagents were supplemented with 1% (*v*/*v*) penicillin-streptomycin hereafter. Sections (average size 2 × 2 cm) were initially rinsed with HBSS before encapsulated in an artificial pleura (sodium alginate and calcium chloride) to allow for agarose inflation. Briefly, 2% agarose was slowly injected into the sample and allowed to set whilst on ice. The tissue was further sectioned to accommodate embedding moulds, filled with 4% agarose. Sections were cut into 400 μm slices using a Leica VT1200 s vibratome and cultured in 15 cm Petri dishes, in OptiMem at 37 °C (5% CO_2_).

### 2.15. Immunostaining of Human Lung PCLS

Slices were first fixed with 4% paraformaldehyde and then incubated in sodium borohydride (1 mg/mL) for 15 min to counteract autofluorescence. After washing in PBS (3 × 5 min at room temperature, RT), slices were permeabilised, blocked, and incubated with primary and secondary antibodies using the same protocol as in [49]. A total of 0.5% of Triton X in PBS was used to permeabilise it for 1 h, followed by blocking with 10% Normal Goat Serum (NGS) in PBS. Primary antibodies were added to the slices in 1% (*w*/*v*) Bovine Serum Albumin (BSA), 0.2% Triton-X, and 5% NGS and incubated overnight at 4 °C, on a plate rocker. Washes were with 0.5% BSA and 0.2% Triton X (2 × 30 min at RT), secondary antibodies (as well as DAPI and Phalloidin) were added to the slices in 5% NGS and 0.2% Triton-X, and they were incubated overnight at 4 °C on a plate rocker. Washes were re-performed twice, along with a final wash of PBS (3 × 30 min at RT in total). Slices were then mounted with Fluorosave onto glass slides with tape spacers to allow for the use of coverslips on a ‘thick’ sample. Images were captured using an in-house Nikon Eclipse Ti2 spinning disc confocal microscope.

### 2.16. Statistical Analysis

Data values are expressed as mean ± standard error of mean (S.E.M) of at least three experiments, unless otherwise stated. All statistical tests were carried out using GraphPad Prism, version 9. Student’s *t*-test was used for comparing two groups for statistical analysis. One- or two-way analysis of variance (ANOVA) with Tukey’s post hoc were used for multiple comparisons. Statistically significant values were taken as *p* < 0.05.

## 3. Results

### 3.1. B7-H3 Localisation Differs between Lung Epithelial and Cancer Cells

B7-H3 has been studied in the context of lung cancer cell–immune cell interactions, but the intrinsic roles for this receptor in normal lung epithelial cells compared to lung cancer remain unclear. To assess this, we firstly analysed the protein expression levels and localisation in normal human lung epithelium (16HBE) and two human lung adenocarcinoma cell lines (A549 and H358). The levels of B7-H3 were higher in the cancer cell lines compared to the normal cell lines (Appendix A) and the protein was highly glycosylated in all three cell types (Appendix A), in agreement with previous studies [50]. Images of endogenous B7-H3 revealed that this receptor was predominantly localised to cell–cell adhesions in both carcinoma cell lines, but was instead localised to intracellular puncta in normal lung epithelial cells, with very weak plasma membrane staining being visible (Figure 1A). To investigate the nature of these structures, 16HBE cells were stained with lysosomal and endosomal markers. These intracellular B7-H3 clusters in 16HBE cells were not associated with endosomal markers (Rab10, Rab11, and EEA1) or lysosomal markers (lysotracker; Appendix A). However, we determined that these B7-H3-positive intracellular structures were strongly colocalised with IMPDH2 (Figure 1B,C), a rate-limiting enzyme involved in metabolic control and GTP synthesis, and predominantly found in structures known as rods and rings (RRs; [51]). 16HBE cells were assembled on average ~2 RRs/cell (Figure 1D), in line with previous reports on other epithelial cells [21]. Images of human lung slices further demonstrated the presence of IMPDH2 and B7-H3 in RR structures in both non-AT2 and AT2 lung epithelial cells (Appendix A), whereby the latter of which are the best characterised origin of lung adenocarcinoma. Conversely, RRs were not seen under basal conditions in either cancer cell line (Figure 1B,D), and IMPDH2 was instead cytoplasmic or localised in the plasma membrane. These data demonstrate that B7-H3 shows distinct localisation in normal lung epithelial cells compared to lung cancer cells and may be associated with IMPDH2.

### 3.2. B7-H3 Negatively Regulates Rod and Ring Assembly in Cancer Cells

We depleted B7-H3 from all three cell lines using either siRNA or CRISPR/Cas9 to determine whether this receptor contributed to IMPDH2 localisation. Successful knockdown was confirmed using Western blot, qPCR, and immunostaining (Appendix A). The knockdown of B7-H3 in 16HBE cells led to a significant increase in RR formation (Figure 2A,B), and when using two individual siRNAs (Appendix A). Moreover, the overexpression of B7-H3 in 16HBE cells resulted in the suppression of RR formation (Figure 2C). The treatment of cells with the IMPDH inhibitor MPA resulted in significantly higher RRs/cell (Figure 2A,B), as previously shown in other cell types [23,24,26,52], and this coincided with increased colocalisation between B7-H3 and IMPDH2 in RRs (Appendix A). As MPA also inhibits IMPDH1 activity, we treated cells with IMPDH2 siRNA to determine whether the latter enzyme played a key role in our observed phenotypes (Appendix A). IMPDH2 depletion led to a significant increase in RR formation, as shown previously [21,27], and the recruitment of B7-H3 to RRs, as seen with MPA treatment (Appendix A). However, the MPA treatment of 16HBE B7-H3 knockdown cells resulted in fewer RRs/cell compared to the control cells treated with MPA, or untreated B7-H3-depleted cells (Figure 2A,B).

To determine whether this role for B7-H3 in RR/IMPDH2 function was unique to normal lung epithelial cells, we analysed RR formation in control and B7-H3 CRISPR A549 and A358 cells. Neither cancer cell line assembled RRs under basal conditions (as shown in Figure 1), but B7-H3 depletion led to a significant increase in RR assembly (Figure 2D,E). The re-expression of B7-H3 in A549 CRISPR cells reduced RRs back to levels seen in the control cells, demonstrating the specificity of the CRISPR targeting (Appendix A). MPA treatment induced RR formation to a similar degree in parental and B7-H3 CRISPR cells (Figure 2D,E), and induced B7-H3 recruitment to RRs (Appendix A). This is in contrast with 16HBE cells, which showed fewer RRs/cell in MPA-treated B7-H3 knockdown cells compared to MPA-treated controls. Interestingly, MPA treatment also resulted in a significant reduction in plasma-membrane-associated IMPDH2 in A549 and H358 cells (Figure 2F,G), suggesting that this may represent the key active pool of IMPDH2 in cancer cells. These findings collectively demonstrate that B7-H3 suppresses RR formation under basal conditions. However, B7-H3 depletion reduces the formation of IMPDH2 containing RRs in the presence of MPA in cancer cells, indicating that B7-H3 plays a key role in the control of RRs, and therefore potentially in IMPDH2 activation status, under stress conditions.

### 3.3. B7-H3 Forms a Complex with IMPDH2 and Regulates Downstream Metabolic Targets

Given that B7-H3 colocalised with IMPDH2 in RRs in epithelial cells and in the membrane in cancer cells, we next sought to determine whether these two proteins form part of the same biochemical complex. The immunoprecipitation of endogenous B7-H3 from cells with or without MPA treatment revealed the presence of IMPDH2 in all of the complexes (Figure 3A). Significantly higher IMPDH2 was seen in the complex with B7-H3 in 16HBE cells treated with MPA compared to controls (Figure 3B). We note that the diffuse and apparent differences in the molecular weight of B7-H3 are likely due to an altered glycosylation status (Appendix A). The immunoprecipitation of endogenous IMPDH2 from cells further revealed a similar trend of complex formation between these two proteins in all of the cells (Figure 3C).

To determine whether this complex contributed to IMPDH2-dependent metabolism, we performed mass spectrometry analysis of the lysates from the control cells depleted of B7-H3 or treated with MPA to assess metabolic changes. Several key metabolites were altered in B7-H3-depleted cells (Appendix A), including GTP, which was significantly reduced in all of the cells upon B7-H3 depletion or MPA treatment (Figure 4A). GTP is a key product of IMPDH2 activation, suggesting the removal of B7-H3 which indeed suppresses the activity of IMPDH2. GSSG is a metabolite involved in glutathione-dependent metabolism and its ratio with GSH is considered to be an indicator of oxidative stress [53]. A significant reduction in GSH:GSSG ratios were seen in all of the samples upon the depletion of B7-H3 or treatment with MPA (Figure 4B). We also noted lower basal GSH:GSSH ratios in both cancer cell lines compared to 16HBE normal epithelial cells, in agreement with previous reports showing dysregulated oxidative stress in cancer. These data collectively indicate that B7-H3 could activate IMPDH2, potentially through the formation of a complex with this molecule, and this leads to elevated GTP levels and the maintenance of glutathione-dependent metabolism for the suppression of oxidative stress and increased stress tolerance.

### 3.4. B7-H3 Suppresses Oxidative Stress and Promotes Chemoresistance and Cell Survival

Some chemotherapeutic agents induce oxidative stress to drive cancer cell death. Increasing ROS production or inhibiting antioxidants can reduce cancer cell growth [29]. The excess levels of ROS cause damage to vital cellular pathways, which can lead to the activation of cell death processes such as apoptosis. Given the observed B7-H3/IMPDH2-dependent effects on glutathione-dependent metabolism and GTP, we sought to determine whether these molecules may co-operate to enable cancer cell chemoresistance. The depletion of B7-H3 from all three cell lines resulted in increased ROS and oxidative stress (as measured using CellROX fluorescent reporters in live cells), an effect that was enhanced upon serum starvation in 16HBE cells, but not in A549 or H358 carcinoma cells (Figure 5A). To determine whether chemotherapy-induced toxicity to cancer cells was B7-H3-dependent, we analysed caspase activation using a live cell reporter probe. The enhanced oxidative stress seen in the A549 and H358 cells correlated with increased levels of caspase cleavage (and therefore presumed apoptosis) and reduced cell numbers (Figure 5B,C). Interestingly, B7-H3 depletion also led to significantly higher caspase activation in both lung cancer cell lines in response to serum starvation or cisplatin treatment (Figure 5B,C), suggesting that B7-H3 can protect cells from stress-induced cancer cell death.

To determine whether IMPDH2 contributed to these B7-H3-dependent phenotypes, experiments were repeated in the presence of MPA to block IMPDH activity. MPA increased oxidative stress (Figure 6A) and apoptosis (Figure 6B,C) in control cells (Figure 6A–C), and this effect was significantly enhanced in B7-H3-depleted cells compared with parental controls (Figure 6A–C). These data demonstrate that B7-H3 may act to protect cells from MPA-induced toxicity through stabilising the active form of IMPDH2. Collectively, these findings support a role for B7-H3 in promoting IMPDH2 activity, chemoresistance, and survival via the suppression of oxidative stress and subsequent apoptosis.

### 3.5. IMPDH2 Supports B7-H3-Dependent Cancer Cell 3D Invasion and Proliferation

We next aimed to determine whether B7-H3 may contribute to tumorigenic phenotypes through the control of IMPDH2. B7-H3 depletion in all of the cell lines did not lead to any changes in proliferation, cell spread area or cell shape of cells on 2D surfaces (Appendix A). Moreover, live time lapse imaging of cells for 16 h and subsequent tracking showed no significant changes to migration speed or distance in any cell line (Appendix A).

To test this in more physiologically relevant conditions, we generated spheroids of control of B7-H3-depleted A549 and H358 cells, embedded them in collagen I hydrogels and assessed proliferation and spheroid expansion after 24 h of culture. The quantification of the resulting confocal images (Figure 7A) revealed a significant increase in expansion of A549 and H358 B7-H3 CRISPR cell spheroids compared to parental controls, with increased apparent invasion as evidenced by single cells detaching from the spheroid mass (Figure 7A,B). This increased spheroid expansion was also accompanied by increased growth of B7-H3-depleted cells within the spheroid compared to parental controls (Figure 7C). The incubation of spheroids with MPA resulted in a significant reduction in both expansion and proliferation in B7-H3 knockdown cells with reduced free invading cells at the spheroid periphery, with no effect on control cells (Figure 7A–C), suggesting that IMPDH positively regulates these phenotypes in the absence of B7-H3. Images of spheroids stained for F-actin revealed a loss of cell–cell adhesion integrity at the periphery of B7-H3 CRISPR cells and an apparent release of invasive cells, suggesting that a loss of B7-H3 promotes cell–cell junction destabilisation and enhances cell–ECM adhesion to support the observed phenotypes (Appendix A).

To further explore the relationship between spheroid phenotypes and IMPDH2, H358 parental and CRISPR spheroids were stained for IMPDH2 and F-actin (Figure 7D). Quantification confirmed a significant reduction in F-actin at cell–cell junctions in B7-H3-depleted cells at the periphery of spheroids, which was partially restored upon MPA treatment (Figure 7E, left graph). IMPDH2 was predominantly at the periphery of control cells, with IMPDH2-containing RRs seen in peripheral B7-H3 CRISPR cells. MPA treatment significantly increased RR formation in all of the cells (Figure 7E, right graph), in contrast to cells in 2D where MPA had no additive effect on RR assembly (Figure 2E). These findings suggest that a loss of B7-H3 initiates RR assembly and cell–cell junction disassembly in cells in contact with the ECM, leading to a dominant phenotype in 3D. The full inhibition of IMPDH promoted RR assembly in all the cells, reducing the functional effects seen in B7-HE-depleted cells.

To further explore how B7-H3 depletion leads to the suppression of actin assembly between cells, we analysed the formation of cell–cell adhesions in all of the cell lines in 2D. The depletion of B7-H3 did not alter the localisation of the key adherens junction β-catenin (Appendix A). However, levels of the tight junction protein ZO-1 were significantly reduced in cancer cell lines following B7-H3 removal (Figure 8A,B), and this resulted in significant reductions in ZO-1 localisation at cell–cell adhesions (Figure 8C,D). Collectively, these findings demonstrate that depleting B7-H3 leads to the destabilisation of tight junctions, and this results in enhanced lung adenocarcinoma cell 3D growth and spheroid expansion.

## 4. Discussion

Our study provides new evidence that B7-H3 promotes viability and lung epithelial homeostasis through the maintenance of cell–cell adhesion and regulation of IMPDH2 activity. We propose that B7-H3, through association with IMPDH2, can protect cells against oxidative stress and influence purine metabolism, leading to the modulation of tumour growth and invasion. These effects were enhanced upon B7-H3 knockdown in 3D spheroids, resulting in increased growth and expansion/invasion potentially through protection against oxidative stress and the disassembly of tight junctions.

Several previous investigations of B7-H3 have focussed on tumour cells; however, the localisation and role of B7-H3 in normal epithelia versus cancer cells have not been explored. We show that the expression of B7-H3 is lower in normal lung epithelial cells compared to cancer cells, which agrees with previous findings [54]. We further reveal that B7-H3 localises predominantly to RRs in normal epithelia in vitro and in human lung tissue, but predominantly localises to the plasma membrane and cell–cell junctions in cancer cells. Moreover, increased B7-H3 levels are correlated with increased IMPDH2 membrane localisation, increased GTP synthesis, and enhanced protection against oxidative stress and toxin-induced apoptosis in parental cells. Interestingly, IMPDH2-containing RRs do not assemble under basal conditions in cancer cells, indicating the enhanced activity of this enzyme to promote cell proliferation and invasion, as has been previously suggested [55]. The data that we provide further suggest that the B7-H3:IMPDH2 complex spatially regulates this enzyme. Given the lack of tools to measure IMPDH2 activity, we are unable to assign this specifically to spatial changes in IMPDH2 activity per se. We also note that the ligand or extracellular binding partner for B7-H3 remains unknown. It is plausible that this ligand is present on the surface of adjacent cancer cells, but not normal lung epithelial cells, and this results in the altered localisation observed, and possibly also an altered B7-H3 function and conformational state to regulate IMPDH2 binding. Our data strongly imply that B7-H3 promotes IMPDH2 activation, as evidenced by (1) reduced GTP levels and (2) the induction of RRs upon B7-H3 depletion, which mimics that seen upon the inhibition of IMPDH2 activity using MPA. Thus, we would speculate that B7-H3 regulates the localisation, and potentially activation, of IMPDH2 in the plasma membrane of cancer cells, not normal epithelial cells, which may result in the locally available GTP pools required for the rapid activation of GTPases and potentially a response to external stress. In support of this, IMPDH2 has recently been shown to localise to the plasma membrane of cancer cells and control local GTP synthesis to support actin reorganisation [56].

B7-H3KD cells in 3D showed increased proliferation, invasion, and a dependence on IMPDH2 activity for these phenotypes, as evidenced by the reduction following MPA treatment. The reduced F-actin integrity at the periphery of CRISPR cell spheroids indicates that B7-H3 is required for the stabilisation of these structures. We speculate that this is either through the association of this receptor with an unknown ligand on the surface of adjacent cells, and/or due to reduced GTP levels in these cells that would be expected to compromise the activation of Rho GTPases that are required for F-actin organisation. However, our data would suggest that B7-H3 depletion is not sufficient to relocalise IMPDH2 to RRs in 3D spheroids to the extent seen in 2D cultures. Moreover, the enhanced RR assembly in B7-H3-depleted cells was restricted to those at the edge of the spheroids, in contact with the ECM. This may indicate that B7-H3 plays a dominant role in cells in contact with the ECM, and raises an interesting possibility that the matrix, or intracellular signals arising from adhesion, act in synergy with B7-h3 to control proliferation and migration.

Cells in 3D can exhibit different signalling and metabolic reprogramming, due to the altered topography and organisation of the ECM, and greatly enhanced cell–cell contact compared to those in 2D [57]. B7-H3 depletion in 3D resulted in phenotypes that differ to some previous in vivo studies, which showed pro-tumorigenic roles for B7-H3 [58,59,60,61]. However, many of these studies focussed on other cancer types and used immunocompetent mice, where the interplay between B7-H3 and the immune compartment may influence B7-H3 function. In xenograft or immunodeficient mouse models, the presence of a vascular system and mechanical shear/forces may also contribute to B7-H3 roles in context. In 3D spheroid models, cells are predominantly in contact with each other with no immune cells present, which is more reflective of the environment seen in ‘immune cold’ tumours. We propose that B7-H3 depletion in these types of tumours might drive pro-tumorigenic activity through downstream IMPDH2 activation and the destabilisation of ZO-1 at cell–cell adhesions to promote growth and invasion, as has been suggested in other cancer types [62].

Our data demonstrate that inhibiting IMPDH using MPA alone had no significant effect on 3D growth or invasion in parental cells, indicating a role for IMPDH2 in tumorigenesis only in the context of drug resistance or increased oxidative stress. Chemoresistance to 5-FU in colorectal cancer has been shown to be induced by B7-H3 through enhanced PI3K/AKT signalling, but the role of IMPDH2 in this system has not been explored [63]. B7-H3 expression has also been correlated with paclitaxel resistance in breast cancer, where B7-H3 knockdown increased drug sensitivity [64]. It would be interesting to further explore the mechanisms through which B7-H3 contributes to drug resistance and whether IMPDH2 plays a role in these pro-tumorigenic phenotypes in 3D in B7-H3KD cells, as the 2D data suggest that a loss of B7-H3 would sensitise cells to chemotherapy. We also note that both the depletion of B7-H3 and treatment with MPA elicited reductions in both GTP and the GSH:GSSG ratio coupled with enhanced oxidative stress, suggesting a dependence on IMPDH downstream of B7-H3. Our data do not allow for conclusions to be drawn regarding interactions between GTP and GSH:GSSH/oxidative stress, nor the discrimination of the direction of interaction between these signalling events, should one exist. However, some relationships between these pathways have been described in the literature previously. One study showed that the active GTPases Rit (which requires GTP for activation) acts as a pro-survival molecule in response to oxidative stress [65], indicating that GTP depletion may impair this pathway, placing GTP upstream of an efficient stress response. A recent paper also demonstrated that IMPDH2 and GTP are required for the modulation of GSH:GSSG-driven stress response resolution in stem cells, acting downstream of p38a [66]. At present, we can only conclude that GTP and GSH:GSSG levels/oxidative stress were correlated in our experiments. Future studies to determine whether a direct relationship exists between IMPDH2, GTP, and oxidative stress will be required to understand how these pathways might converge in cancer cell stress responses.

The data presented here suggest that in the absence of immune cells, B7-H3 could maintain cancer cell growth by regulating IMPDH2 localisation and potential activity, and subsequently the de novo purine biosynthetic pathway, resulting in increased GTP levels. This would be expected to activate GTPases, increasing both proliferation and invasion at the periphery of the tumour and protecting cells against oxidative stress in 3D. Depleting B7-H3 reduces IMPDH2 activation and downstream targets as well as increases IMPDH-2-dependent proliferation and invasion in 3D spheroids. It would be interesting to determine in future studies whether B7-H3 forms a direct complex with IMPDH2 and to manipulate this complex directly to explore tumour cell behaviour in a 3D context. Indeed, as the ligand for B7-H3 remains elusive, defining non-immune-related extracellular binding partners for B7-H3 in normal and cancerous tissues may aide in determining how B7-H3 subcellular localisation is controlled. Interestingly, PD-L1 has been reported to mediate cancer cell intrinsic functions in cold tumours through mTOR activation [67], and IMPDH2 promotes proliferation and migration through mTOR activation, suggested to be downstream of B7-H3 [47,68,69]. Future experiments to investigate the involvement of mTOR signalling in B7-H3-dependent oxidative stress and roles in DNA damage response pathways would be important to understand the potential signalling consequences of B7-H3 inhibition or depletion [70]. It is plausible that the B7-H3-dependent suppression of oxidative stress in a 3D environment may suppress DNA damage and genomic instability, thereby leading to the anti-tumorigenic phenotypes seen in our study [70,71].

## 5. Conclusions

In summary, our study provides new evidence that B7-H3 is associated with one of the key rate-limiting metabolic enzymes IMPDH2, and this complex localises to intracellular RRs in normal epithelia and the plasma membrane in cancer cells that overexpress B7-H3. The IMPDH2:B7-H3 complex provides a protective role in cancer cells to promote cell survival through the maintenance of downstream metabolic activity and oxidative stress. Importantly, we demonstrate significant changes in B7-H3 function in cancer cells between traditional 2D and more physiologically relevant 3D culture conditions. In the latter, B7-H3 negatively regulates growth and expansion/invasion, as well as the partial relocalisation of IMPDH2, which may in turn control the balance of GTPase activation required for efficient growth and motility. We propose that B7-H3 plays important non-immunogenic roles in tumour cells to regulate the balance of active and inactive IMPDH2, which is critical to cell survival in response to extracellular stress. Whilst B7-H3 may remain an interesting target for immunotherapy, these findings caution against the use of anti-B7-H3 therapies in certain clinical settings.

## Figures and Tables

**Figure 1 cancers-15-03530-f001:**
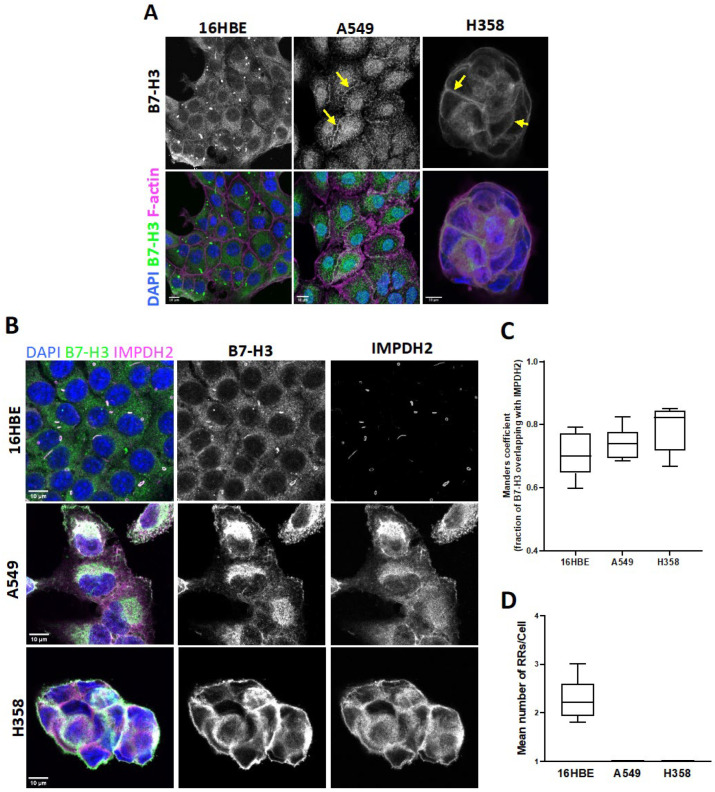
B7-H3 localisation differs between lung epithelial and cancer cells. (**A**) Representative confocal images of 16HBE, A549, and H358 cells stained for endogenous B7-H3 levels (green), F-actin (magenta), and nuclei (blue). Arrows denote localisation to cell-cell adhesions. (**B**) Representative confocal images of cells (**A**) stained for nuclei (blue), B7-H3 (green middle panel), and IMPDH2 (magenta right panel). Scale bar: 10 µm. (**C**) Quantification of colocalisation between B7-H3 and IMPDH2 from images, as in (**B**). Analysis carried out using Manders’ coefficient and represents mean ± SEM from 5 fields of view with at least 10 cells/field of view and from 7 independent experiments. (**D**) Quantification of mean number of RRs in cells (**B**) stained for IMPDH2. Values represent mean ± SEM from 5 fields of view with at least 10 cells/field of view and from 7 independent experiments. Note: no RRs are visible in A549 or H358 cells. Arrows indicate B7-H3 localisation at cell–cell adhesions.

**Figure 2 cancers-15-03530-f002:**
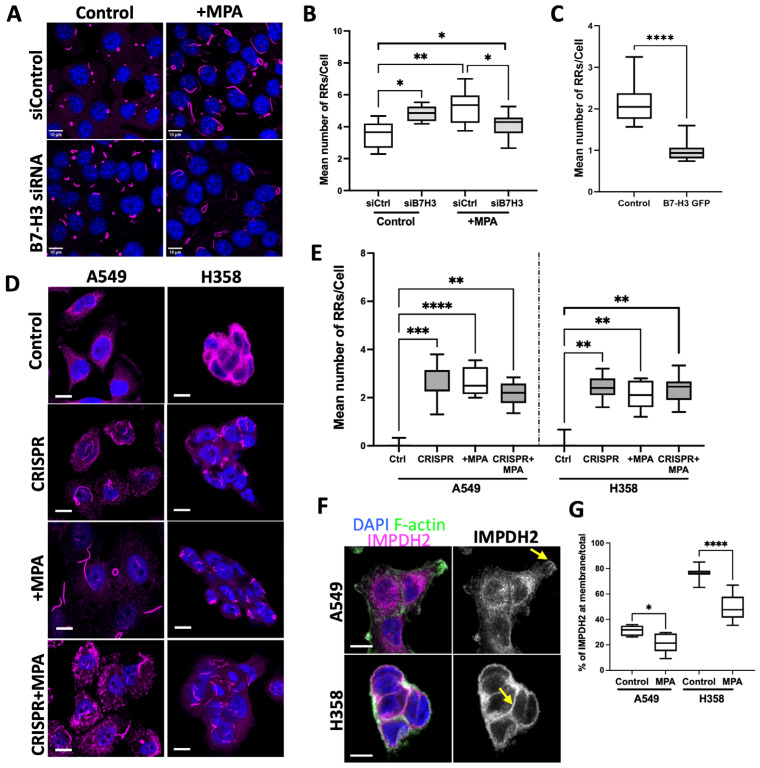
B7-H3 negatively regulates rod and ring assembly in cancer cells. (**A**) Representative confocal images of 16HBE cells transfected with control (non-targeted) siRNA or B7-H3 siRNA, treated with MPA for 48 h, fixed, and stained for DAPI (blue) and IMPDH2 (magenta). (**B**) Quantification of the mean number of RRs per cell from images of control or MPA-treated cells in both siRNA control and siB7-H3 in 16HBE cells. Data are representative of three independent experiments. Seven or more cells from 3–9 fields of view were analysed. (**C**) Quantification of the mean number of RRs per cell from images of control and B7-H3GFP cells stained for IMPDH2. Data were pooled from three independent experiments. Seven or more cells from at least 7 fields of view were analysed. (**D**) Representative confocal images of IMPDH2 (magenta) or nuclei (blue) staining in control or MPA-treated A549 and H358 parental and CRISPR cells. Scale bar: 10 µm. (**E**) Quantification of the mean number of RRs per cell from images of control, CRISPR, and +MPA-treated cells stained for IMPDH2. (**F**) Representative images of A549 and H358 cells stained for IMPDH2 (magenta) and F-actin (green). Arrows denote localisation of IMPDH2 to plasma membrane. Scale bar: 10 µm. (**G**) Quantification of plasma-membrane-associated IMPDH2 as proportion of total IMPDH2 signal from images, as in (**F**). Graphs show data scored from 10 different fields of view, one experiment shown, which is representative of 3 independent experiments. Graphs shown as mean ± SEM. Significance assessed via *t*-test or one-way ANOVA; * *p* < 0.05, ** *p* < 0.01, *** *p* < 0.001,**** *p* < 0.0001.

**Figure 3 cancers-15-03530-f003:**
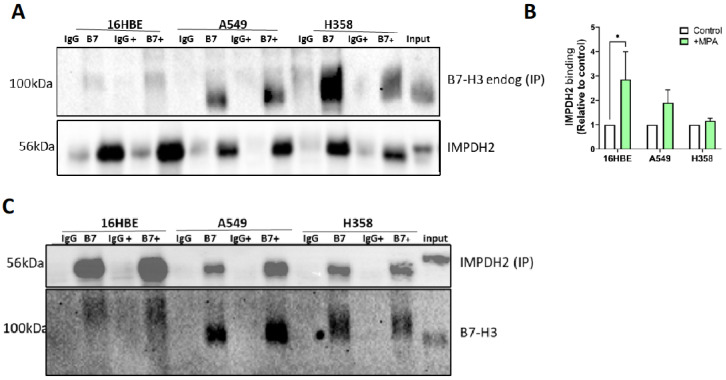
B7-H3 forms a complex with IMPDH2. (**A**) Western blot of endogenous B7-H3 immunoprecipitation from 16HBE, A549, and H358 cells treated with vehicle or MPA (+) for 48 h, incubated with either B7-H3 or IgG control antibody, and then probed for IMPDH2 and B7-H3. (**B**) Quantitative analysis of IMPDH2 levels in B7-H3 endogenous IP, as in (**A**). Data were pooled from 5 independent experiments. Values are plotted as mean ± SEM. (**C**) Western blot of endogenous IMPDH2 immunoprecipitation in cells, as in (**A**), incubated with either IMPDH2 or IgG control antibody followed by immunoblotting for IMPDH2 and B7-H3. One-way ANOVA was used to determine statistical significance. * = *p* < 0.05. The uncropped bolts are shown in Appendix A.

**Figure 4 cancers-15-03530-f004:**
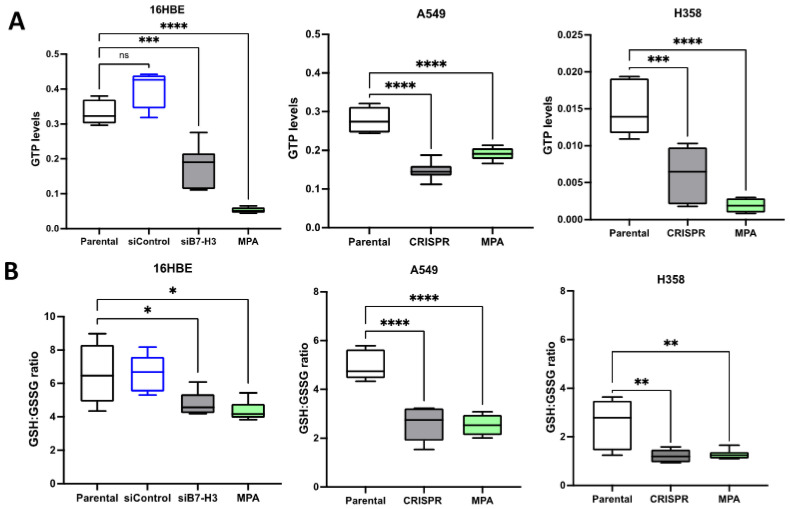
Reducing B7-H3 leads to reduced IMPDH2 pathway metabolites. (**A**) Graph of GTP levels quantified via mass spectrometry in parental/siControl, B7-H3 siRNA/CRISPR knockdown, and MPA treated in 16HBE, A549, and H358 cells. (**B**) Quantification of GSH:GSSG ratio in cells, as in (**A**). n = 6 samples per condition from 2 independent experiments. Values are plotted as mean ± SEM. One-way ANOVA was used to determine statistical significance. * = *p* < 0.05, ** = *p* < 0.01, *** = *p* < 0.001, **** = *p* < 0.0001.

**Figure 5 cancers-15-03530-f005:**
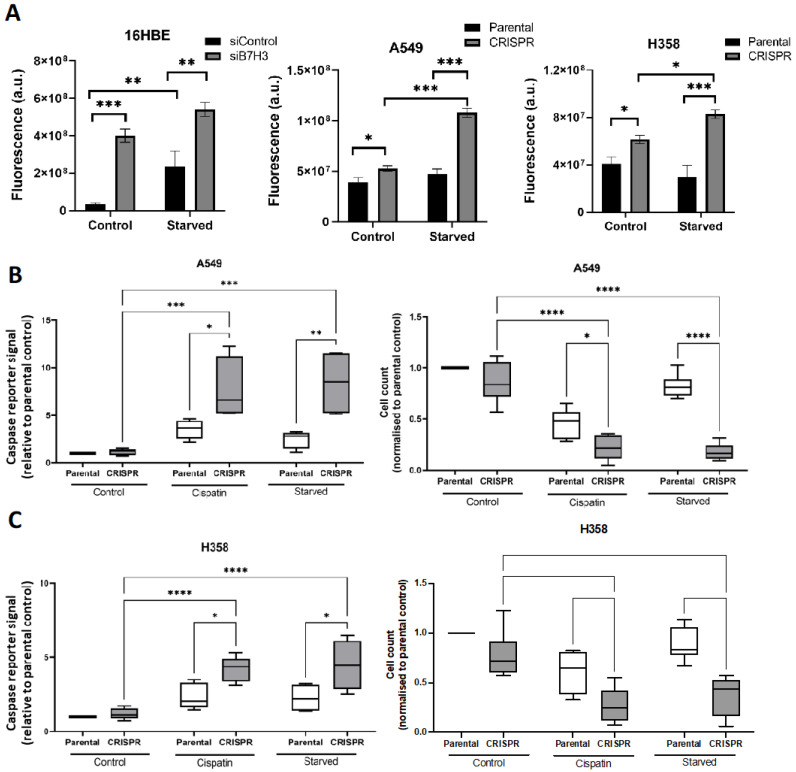
B7-H3 suppresses oxidative stress and promotes chemoresistance and survival. (**A**) Quantification of oxidative stress levels by measuring the CellROX fluorescence signal under specified conditions. (**B**,**C**) Graphs of apoptosis (left panel) and cell count (right panel) in parental or B7-H3 CRISPR A549 cells (**B**) or H358 cells (**C**). Cells were treated with DMSO as the vehicle control, or were cisplatin- or serum-starved for 48 h prior to staining the live cells with DAPI and CellEvent™ Caspase-3/7 Green detection reagent for apoptosis. Data were pooled from 3 independent experiments, with 3 replicates per experiment. Values are plotted as mean ± SEM. One-way ANOVA was used to determine statistical significance. * = *p* < 0.05, ** = *p* < 0.01, *** = *p* < 0.001, **** = *p* < 0.0001.

**Figure 6 cancers-15-03530-f006:**
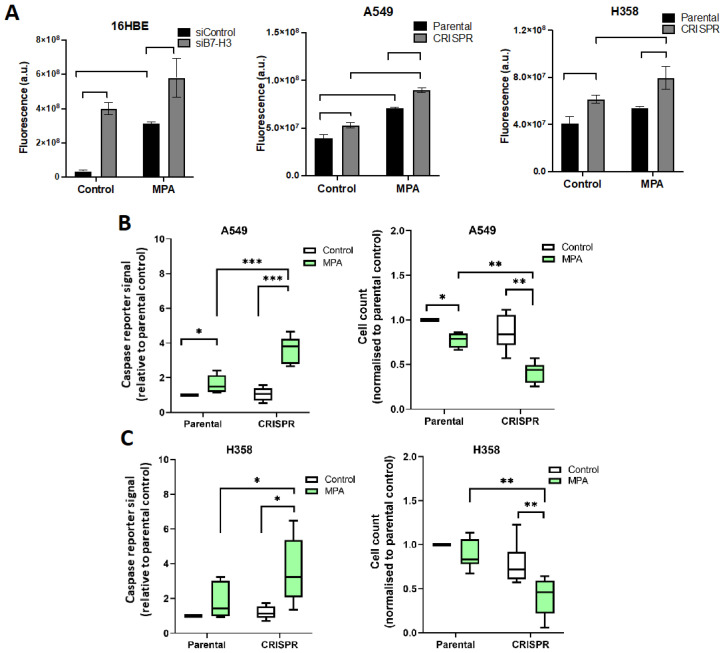
B7-H3 sensitises cells to IMPDH2-dependent oxidative stress, chemoresistance, and survival. (**A**) Quantification of oxidative stress levels of siRNA control/parental or B7-H3 siRNA/CRISPR cells treated with MPA or vehicle control for 48 h prior to incubation with CellROX reagent. Data were pooled from 3 independent experiments; each experiment was performed in triplicate. (**B**,**C**) Graphs of apoptosis and cell numbers in parental or B7-H3 CRISPR A549 (**B**) and H358 (**C**). Cells were treated with methanol as the vehicle control or MPA for 48 h prior to staining the live cells with DAPI and CellEvent™ Caspase-3/7 Green detection reagent to measure apoptosis. Data were pooled from 3 independent experiments; each experiment was performed in triplicate. Values are plotted as mean ± SEM. One-way ANOVA was used to determine statistical significance. * = *p* < 0.05, ** = *p* < 0.01, *** = *p* < 0.001.

**Figure 7 cancers-15-03530-f007:**
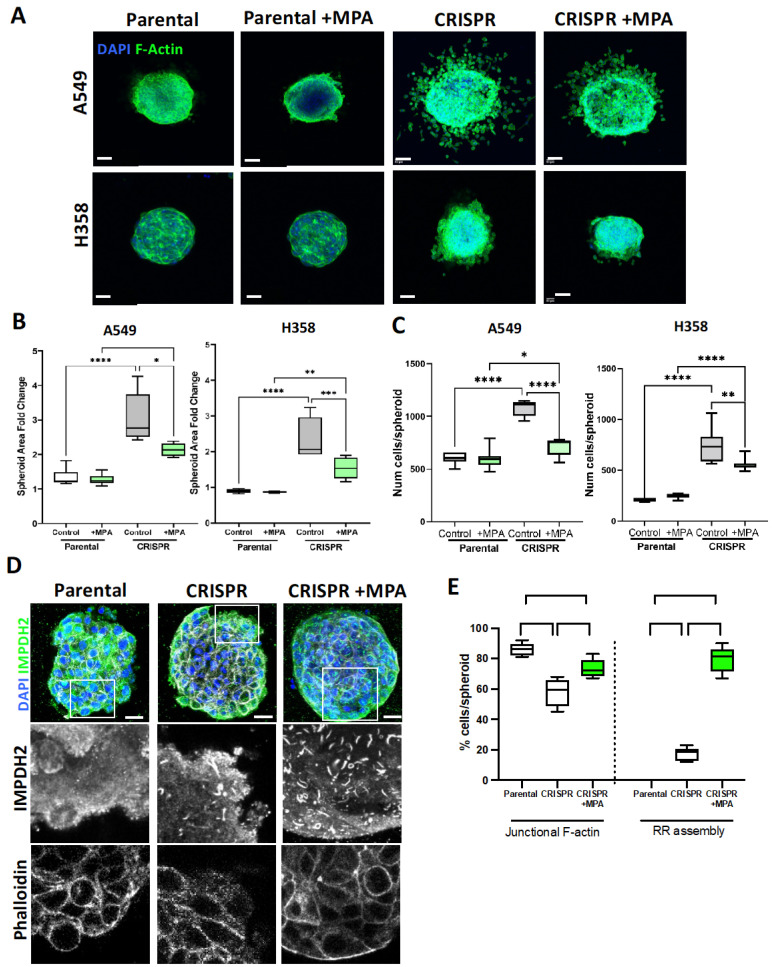
IMPDH2 contributes to B7-H3-dependent 3D invasion and proliferation. (**A**) Representative confocal images of spheroids from parental and B7-H3 CRISPR A549 and H358 cells embedded in collagen, either vehicle- or MPA-treated, fixed, and stained for DAPI (blue) and phalloidin (green) 24 h post-embedding. Scale bar: 50 µm (**B**) Quantification of spheroid area over 24 h post-embedding. (**C**) Quantification of the number of cells per spheroid using QuPath software. (**D**) Example confocal images of H358 parental and B7-H3 CRISPR spheroids either vehicle- or MPA-treated, fixed, and stained for DAPI (blue) and IMPDH2 (green) 24 h post-embedding. Scale bar: 20 µm. IMPDH2 and F-actin shown as inset panels below. (**E**) Quantification of images, as in D, of % of cells/spheroids showing junctional F-actin and R assembly. Data shown in (**B**,**C**,**E**) are pooled from three independent experiments with a total of 6 spheroids analysed per experiment. One-way ANOVA was used to determine statistical significance. * = *p* < 0.05, ** = *p* < 0.01, *** = *p* < 0.001, **** = *p* < 0.0001.

**Figure 8 cancers-15-03530-f008:**
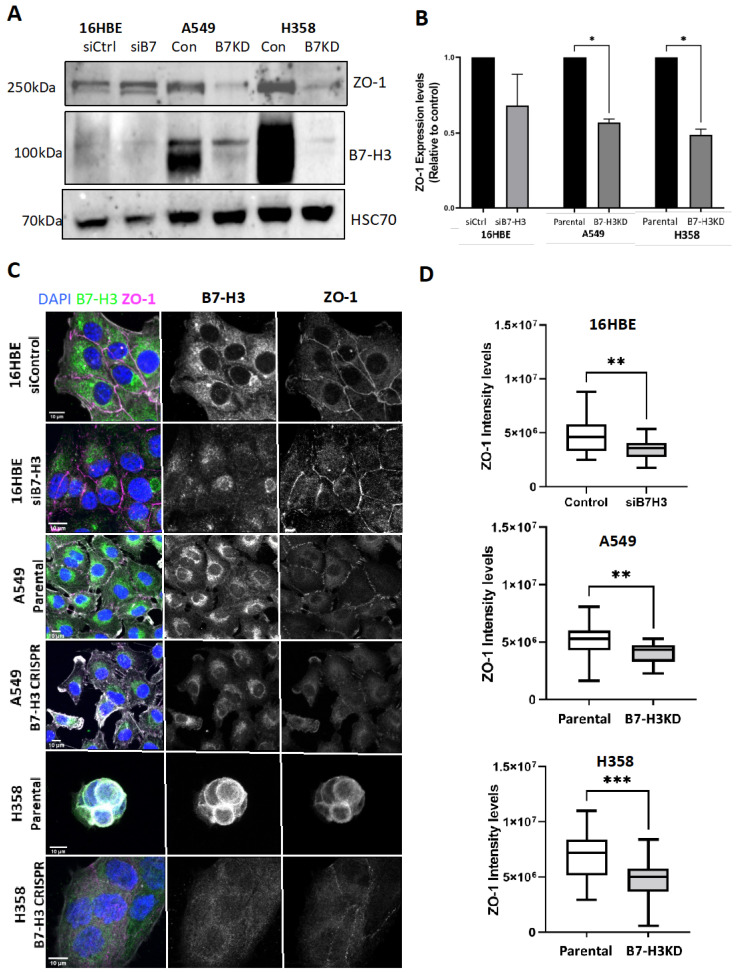
B7-H3 knockdown reduces ZO-1 levels and localisation to tight junctions. (**A**) Western blot of ZO-1 levels in lysates from 16HBE, A549, and H358 control, as well as B7-H3KD cells, probed for ZO-1, B7-H3, and HSC70 (loading control). (**B**) Quantification of data, as in (**A**), from n = 4 experiments. (**C**) Representative confocal images of siRNA control/parental or B7-H3 siRNA/CRISPR cells fixed and stained for nuclei (blue), ZO-1 (magenta), and B7-H3 (green). (**D**) Quantification of fluorescence intensity of ZO-1 from images, as in (**C**). Intensity is shown for at least 50 cells per cell line and 10 fields of view from 3 independent experiments. Data are shown as mean ± SEM; significance was assessed via t-test or one-way ANOVA; * *p* < 0.05, ** = *p* < 0.01, *** = *p* < 0.001. Scale bar: 10 µm. The uncropped bolts are shown in Appendix A.

## Data Availability

The data presented in this study are available upon request from the corresponding author.

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
