# Peer review of "B7-H3 Associates with IMPDH2 and Regulates Cancer Cell Survival"

_cancers, 2023, doi:10.3390/cancers15133530_

Round 1

Reviewer 1 Report (Previous Reviewer 1)

The authors addressed all my reviews

Author Response

We thank the reviewer for evaluating our revised manuscript and for their conclusion that our responses addressed their original comments.

Reviewer 2 Report (Previous Reviewer 3)

This manuscript reports that the authors studied the effect of B7-H3 on cancer cell survival via IMPDH2. In addition, it has been observed that the B7-H3:IMPDH2 complex affects cancer cell survival by regulating oxidative stress in cancer cells. Furthermore, it was observed that the B7-H3 works in a 3D model, unlike a 2D model. When this reviewer first reviewed this manuscript, this reviewer suggested adding additional small animal (mice) experiments and clinical data. The authors responded to the reviewer’s comment. The authors agreed that small animal experiments were necessary and, although not presented in this manuscript, the authors note that "Others have previously depleted B7-H3 from tumour cells or treated animals with MPA and performed in vivo analysis of tumor growth.". Therefore, this reviewer believes that the authors need to perform additional research, such as small animal experiments, to better improve this manuscript.

In Figure 3A, when immunoprecipitation (IP) of BH-H3 was performed, the signal of BH-H3 in 16HBE cells was too weak, but the signal of IMPDH2 was strong. If anti-B7-H3 was used for IP, this reviewer thinks the signal of B7-H3 should be stronger than IMPDH2. In addition, although the amount of B7-H3 expression is different for each cell, this reviewer thinks that the amount of B7-H3 is similar and the amount of IMPDH2 is different when Co-IP is performed. This reviewer thinks the results should be clarified. In addition, there is a difference between immunoprecipitated samples and input size in western blots. This reviewer thinks this part needs to be clarified.

Author Response

Response to Reviewer 2 Comments

Point 1: This manuscript reports that the authors studied the effect of B7-H3 on cancer cell survival via IMPDH2. In addition, it has been observed that the B7-H3:IMPDH2 complex affects cancer cell survival by regulating oxidative stress in cancer cells. Furthermore, it was observed that the B7-H3 works in a 3D model, unlike a 2D model. When this reviewer first reviewed this manuscript, this reviewer suggested adding additional small animal (mice) experiments and clinical data. The authors responded to the reviewer’s comment. The authors agreed that small animal experiments were necessary and, although not presented in this manuscript, the authors note that "Others have previously depleted B7-H3 from tumour cells or treated animals with MPA and performed in vivo analysis of tumor growth". Therefore, this reviewer believes that the authors need to perform additional research, such as small animal experiments, to better improve this manuscript.

Response 1: We thank the reviewer for their comments and perspective. As we noted in our previous response, other studies have depleted B7-H3 from cancer cells and undertaken xenograft experiments using human cells in immune compromised mice, or subcutaneous injections of murine cancer cells into immune competent animals. To contribute relevant new in vivo data to complement existing studies and our in vitro and 3D model experiments in the present study, the binding site between B7-H3 and IMPDH2 would need to be fully mapped, characterised in vitro, analysed in 3D models and the mutant B7-H3 protein reconstituted into B7-H3 null cells for onward injection into animals and metabolic mapping of resulting tumours. We strongly believe that the extent of these studies fall outside the scope of the current manuscript, and instead represent an entirely new study for future exploration. We hope the reviewer will agree with our summary of the proposed experiments and that the relevant in vivo experiments required should remain a goal for future studies.

Point 2: In Figure 3A, when immunoprecipitation (IP) of BH-H3 was performed, the signal of BH-H3 in 16HBE cells was too weak, but the signal of IMPDH2 was strong. If anti-B7-H3 was used for IP, this reviewer thinks the signal of B7-H3 should be stronger than IMPDH2. In addition, although the amount of B7-H3 expression is different for each cell, this reviewer thinks that the amount of B7-H3 is similar and the amount of IMPDH2 is different when Co-IP is performed. This reviewer thinks the results should be clarified. In addition, there is a difference between immunoprecipitated samples and input size in western blots. This reviewer thinks this part needs to be clarified.

Response 2: We thank the reviewer for these observations. We note that it is not possible to compare levels of different proteins on western blots that have been probed with different antibodies and subjected to different exposure times as the affinity and avidity of each antibody is different. As we show in Figure S1, B7-H3 is expressed at much lower levels in 16HBE cells compared to both cancer cells lines, hence the immunoprecipitations (which was performed using the same amount of antibody for each cell line) would be expected to retrieve lower levels of protein in 16HBE cells. We believe the higher relative level of IMPDH2 binding seen in B7-H3 complexes (Figure 3A) and vice versa in IMPDH2 complexes (Figure 3C) is likely due to the enhanced IMPDH2 colocalization seen between these proteins in 16HBE normal lung epithelial cells. With regards to the different molecular weights seen in inputs vs. IP lanes: this was raised in the previous review round, and we responded to this by adding text (lines 510-512 of our previous submission) in our manuscript to state the likely explanation, which relates to the glycosylation status of B7-H3 which may be enriched in the immunoprecipitated samples as compared to the lower total protein input lane. We hope this further clarifies this point for the reviewer.

Reviewer 3 Report (Previous Reviewer 4)

Authors have adequately performed the suggested experiments and have satisfactorily responded to the questions/objections raised.

Author Response

We thank the reviewer for evaluating our revised manuscript and for their conclusion that our responses addressed their original comments.

This manuscript is a resubmission of an earlier submission. The following is a list of the peer review reports and author responses from that submission.

Round 1

Reviewer 1 Report

In this manuscript, Alhamad et al. investigate the interaction of B3-H7 with IMPDH2 in lung cancer. Even if the authors present a large quantity of data and propose an interesting hypothesis, the conclusions are not supported by the results and the presented data leave many questions open. The main working hypothesis is that B3-H7 is required to maintain a basal activity of IMPDH2 by preventing its localization to RRs. However, the relationship between RR localization and activation is not clear. Furthermore, there is a major discrepancy between the data in 2D and in 3D, where B3-H7 seems to have opposite effects on IMPDH2. While it is clear that 3D culture is metabolically different from monolayer culture, it is not clear if all the results presented in 2D apply also for 3D culture, in which the only data with IMPDH2 is an inhibition with MPA, which could have off-target effects. Given the many inconsistencies, missing controls, and conclusions unsupported by solid evidence, this manuscript is not suitable for publication. Following are specific critiques.

The comparison of normal and cancer cell lines raises some concern. The authors use a normal bronchial cell line as normal control; however, the cell of origin of lung adenocarcinoma is not the bronchial epithelium, but type 2 alveolar cells. There are commercially available small airway cell lines which are also mostly made up of ciliated bronchial cells, but contain an alveolar component as well. Alveolar cells would be the appropriate normal control for the two adenocarcinoma cell lines used.

For the siRNA and CRISPR experiments, the results with at least two independent siRNAs/sgRNAs should be shown to exclude off-target effects.

In figure S3C, the control without MPA is not shown.

In figure 2D-E, it is extremely hard to discriminate between cytoplasmic and membrane-associated IMPDH2. How were the authors measure the fraction of membrane IMPDH2 from these pictures??

At lines 412-414, the authors state that “These findings collectively demonstrate that low levels of B7-H3 in normal epithelial cells suppress RR formation, whereas high expression of B7-H3 in cancer cells leads to suppression of RR and IMPDH2 activation”. This conclusion is not supported by the data. The authors show that siRNA targeting B7-H3 increases the localization of IMPDH2 to RRs in basal condition, but not in MPA-treated cells. The authors should comment on this opposite effect of siB7-H3 in basal vs MPA-treated cells. Also, the authors do not show the effect of CRISPR for B7-H3 in cancer cells not treated with MPA, so it is not clear if B7-H3 really inhibits IMPDH2 localization to RR in cancer cells, because this data is not shown anywhere. In the MPA-treated cells in figure 2F, the number of RRs per cell does not seem to be reduced; in fact, the authors express this measurement as % of cells with RR, instead of number of RR per cell, as in figure 2C. Why don’t the authors use consistent measurements for normal and cancer cells?? Finally, nowhere in the results the authors show measurement of IMPDH2 activity, so it is not clear at all why they conclude that high expression of B7-H3 in cancer cells leads to IMPDH2 activation. In the introduction, the authors state that there is no correlation between IMPDH2 activity and localization to RRs, so the functional role of the presented results is not clear at all, and the authors should avoid any reference to IMPDH2 activation, only localization to RRs.

In figure 3A-C, the molecular weight of IMPDH2 and B7-H3 in the input is different from the immunoprecipitated. The authors should comment on this discrepancy.

In lines 457-458, the authors state that “level of GSSG levels were significantly increased in B7-H3 depleted cells”. However, this data is not shown anywhere. Either the authors show the data or they should remove the statement.

In all metabolome experiments (figure S4 and 4) the unit of measure for the metabolites are not shown.

Conceptually, the reason to present the localization data is not clear. The immunofluorescence experiments in figures 1-2 suggest a difference between normal and cancer cells in the localization of B7-H3. However, the functional data shown in the subsequent figures show that B7-H3 has similar function both in normal and in cancer cells, because it similarly interacts with IMPDH2, is required for maintenance of GTP and GSH/GSSG ratio, and its knockdown increases ROS. So what is the purpose of showing the different localizations if the function does not change?

 In figure 5B, both left and right panel are labelled “apoptosis”. First of all, the label “apoptosis” is not correct, because the measurement of a caspase reporter does not equal apoptosis. Second, I believe the right panel should be “cell count”, as in figure 5C.

In figure 5C, the significance is not shown in the right panel.

In figure 5, the effect of siB7-H3 on response to chemotherapy in the normal cell line is not shown.

In lines 500-502, the authors state that “Collectively these findings support a role for or B7-H3 in promoting chemoresistance and survival by suppression of oxidative stress and subsequent apoptosis through stabilising IMPDH2 activity”. This conclusion is not supported by the data. If the anti-apoptotic activity of B7-H3 was due to stabilization of IMPDH2, further inhibition of IMPDH2 activity with MPA would not cause further increase in ROS and apoptosis. The results in figure 6 suggest that the effect of MPA is independent on the effect of B3-H7 siRNA. Also, there is not proof that MPA acts via IMPDH2, since it may have off-target effects. It would be more convincing to show a rescue of ROS and apoptosis by over-expression of IMPDH2 in cells where B7-H3 has been depleted by siRNa or CRISPR.

In lines 522-524, the authors state that “Quantification of resulting 522 confocal images (Figure 7A) revealed a significant increase in invasion of A549 and H358 B7-H3 CRISPR cells compared to parental controls (Figure 7A, B).” The representative images suggest that the CRISPR cells invade more the surrounding collagen. However, the measurement in figure 7B is “spheroid area fold change”. How do the authors use this as a measure of invasion?? Spheroid area and number of cells per spheroids are not a measurement of invasion.

In figure 7 and 8, the authors suggest that B3-H7 depletion reduces cell adhesion and increases invasion, but it is not clear how this part of the story matches the previous part, with the role of B3-H7 in IMPDH2 localization and activity. The authors hypothesize that B3-H7 promotes maintenance of cell-cell adhesion via activation of IMPDH2 and protection against ROS. It is not clear how this is connected to the effect on invasion. In cells lacking B3-H7, IMPDH2 inhibition with MPA seems to reduce the tumor spheroid invasion, so IMPDH2 activity is supposedly required for invasion? This effect is the opposite of the effect observed on “apoptosis” and cell count, where MPA increased caspase activation and reduced cell count. This is very confusing and suggest an off-target effect of MPA.

In lines 574-576, the authors state “RR assembly was only increased in normal lung epithelial cells following B7-H3 knockdown, indicating an increase in the inactive form of IMPDH2 under basal conditions in cancer cells compared to normal lung epithelia.” First of all, the authors never show the localization of IMPDH2 in cancer cells in basal conditions, but only in MPA-treated cells, so they cannot compare the induction of RR assembly in normal vs cancer cells. Second, the role of RR is not clear. In figure 2A, the authors show that B7-H3 siRNA reduces the IMPDH2 localization to RRs. Does it mean that B7-H3 is required for repression of IMPDH2 activity by MPA? This is the opposite of what happens in basal conditions, where B7-H3 seems to be required to keep IMPDH2 out of the RRs. This discrepancy should be addressed. Third, in the introduction the authors state that there is no correlation between IMPDH2 localization to RR and its activity, but throughout the manuscript they present the RR localization as synonymous of inhibition. This discrepancy should be addressed.

The metabolomic and 2D experiments suggest that B3-H7 is required for maintenance of IMPDH2 activity. However, in 3D, MPA attenuates the increased invasion caused by B3-H7 activation. The authors conclude in lines 578-580 that “B7-H3KD in 3D led to increased lung cancer cell proliferation, invasion and IMPDH2 activity as evidenced by the reduction in these phenotypes following MPA treatment”. However, there is no indication that IMPDH2 activity is increased in B3-H7 depleted cells in 3D. This is likely an off-target effect of MPA. Does knockdown of IMPDH2 have the same effect in 3D?

There is no mention of the mechanism by which increased GTP levels would improve the GSH/GSSG ratio and protect against ROS.

It is not clear how B3-H7 would activate IMPDH2 in 2D and inhibit the same enzyme in 3D.

Reviewer 2 Report

Here the authors test a hypothesis that a B7 family receptor B7-H3 (CD276) when complexed with a rate-limiting metabolic enzyme IMPDH2 protects cancer cells from chemotherapy-induced oxidative stress. B7-H3 is often overexpressed in several cancers and is associated with poor treatment outcomes and survival. B7-H3 regulates tumor epithelial-mesenchymal transition pathways, and its down-regulation results in diminished glycolytic capacity and elevated sensitivity to chemotherapeutics. IMPDH2 is also highly expressed in cancer cells and is upregulated during proliferation, and mediates chemoresistance. As IMPDH2 is known to assemble into rods and rings (RRs) micron scale linear or circular intracellular structures, authors first show that both the B7-H3 and IMPDH2 co-localized in RR in lung epithelial cells and at the membranes in the lung cancer cells, followed by co-immunoprecipitation studies to confirm interaction of the endogenous B7-H3 and IMPDH2 proteins. Depletion of B7-H3 results in lower levels of GTP, a IMPDH2-regulated metabolite, and GSH:GSSG ratio in lung epithelial and cancer cells with consequent elevated ROS and oxidative stress and higher apoptosis in response to serum starvation or Cisplatin treatment. These findings suggest role(s) of B7-H3/IMPDH2 complex in promoting cell survival, apoptosis inhibition, and chemoresistance. Importantly, authors show that B7-H3 signaling, in part through interactions with IMPDH2, suppress 3D invasion and proliferation without affecting cell proliferation in 2D assays. Suppression of B7-H3-dependent 3D invasion and proliferation was transduced in part by upregulation of tight-junction protein ZO-1. The authors conclude that although B7-H3 interaction with IMPDH2 contributes to diminished ROS and oxidative stress, and promote cell survival, reduced apoptosis and chemo-resistance, the signaling by this receptor also suppresses 3D invasion and proliferation of cancer cells in part by upregulating tight-junction protein ZO-1. These studies provide evidence of different, context-dependent role(s) of B7-H3 receptor signaling in lung epithelial and cancer cell models, and offer a caution against the use of anti-B7-H3 targeting in clinic for lung and possibly other cancer management.   

Overall the studies are well conducted with appropriate controls and statistical analyses where necessary. Data interpretation is reasonable and consistent with prior observations of B7-H3 signaling and roles in cancer cell proliferation, survival, and drug-resistance.

Editorial comments:

1.       Please also incorporate the caution statement from line 635 into the end of abstract at line 32.

2.       Line 148, NheI and KpnI; Line 157, Essential; Line 485, or;

3.       Figure 3A, why B7-H3 western blot signal is diffuse.

4.       Figure 4B, right plot, Y-axis label: Cell Count.

5.       Figure 8, siB7 lane for 16HBE do not show any reduction in B7-H3 signal. Please correct. Confocal data in 8C is more convincing of depletion/reduction of B7-H3 in HBE cells.

Reviewer 3 Report

In this paper, the authors observed that B7-H3 is involved in cancer cell survival through IMPDH2. The authors identified that B7-H3 is related to and interacts with IMPDH2. In addition, it was proved that the B7-H3:IMPDH2 complex has an effect on cancer cell survival by controlling the oxidative stress of cancer cells. Furthermore, it was identified that B7-H3 works in 3d model unlike 2d model. Overall, the research is robust and well-done. However, studies were conducted at the cellular level to prove the authors' hypotheses. It is considered necessary to carry out additional research such as small animal experiments and clinical data analysis. As a whole, this manuscript needs further experiments.

Reviewer 4 Report

The manuscript “B7-H3 associates with IMPDH2 to regulate cancer cell survival” by Alhamad et al, analyze the role of B7-H3 (CD276), a member of the B7 family of cell surface receptors, in normal lung epithelia, due to its potential role in modulating immune cell responses within the tumour microenvironment.

There are some aspects of the manuscript to improve:

-          Authors analyze how B7-H3 function may differ in normal lung epithelial cells compared with lung adenocarcinoma cells; however, lung adenocarcinomas arise from alveolar epithelial cells, and the 16HBE cells are human bronchial epithelial cells, not alveolar cells. Therefore, the authors should study, at least, the expression by IF of B7-H3 and IMPDH2 in normal lungs (mouse or human) or in normal alveolar cells, to know if the expression in alveolar cells is similar to that of bronchial epithelial cells.

-          Figure 1A. Show images at similar augmentation. Include arrows to show the localization of B7-H3 receptor to cell-cell adhesions in cancer cells.

-Explain the relevance of analyzing the association of intracellular B7-H3 374 clusters in 16HBE cells with lysosomal markers.

-          Discuss the biological relevance of these studies, in cancer cell lines, compared to those in an immunocompetent environment, since the final objective is the applications of the studies in cancer patients, with immune system.